# DS-Trans: A 3D Object Detection Method Based on a Deformable Spatiotemporal Transformer for Autonomous Vehicles

Yuan Zhu [1], Ruidong Xu [1], Chongben Tao [2], Hao An [1], Huaide Wang [1], Zhipeng Sun [3] and Ke Lu [1,*]

[1] School of Automotive Studies, Tongji University, Shanghai 201800, China; yuan.zhu@tongji.edu.cn (Y.Z.); rd_xu@tongji.edu.cn (R.X.); anhao420@tongji.edu.cn (H.A.); huaide_wang@tongji.edu.cn (H.W.)
[2] School of Electronic and Information Engineering, Suzhou University of Science and Technology, Suzhou 215009, China; tom1tao@163.com
[3] Nanchang Automotive Institute of Intelligence and New Energy, Tongji University, Nanchang 330010, China; sunzhipeng@naiine.com
* Correspondence: luke@tongji.edu.cn

**Abstract:** Facing the significant challenge of 3D object detection in complex weather conditions and road environments, existing algorithms based on single-frame point cloud data struggle to achieve desirable results. These methods typically focus on spatial relationships within a single frame, overlooking the semantic correlations and spatiotemporal continuity between consecutive frames. This leads to discontinuities and abrupt changes in the detection outcomes. To address this issue, this paper proposes a multi-frame 3D object detection algorithm based on a deformable spatiotemporal Transformer. Specifically, a deformable cross-scale Transformer module is devised, incorporating a multi-scale offset mechanism that non-uniformly samples features at different scales, enhancing the spatial information aggregation capability of the output features. Simultaneously, to address the issue of feature misalignment during multi-frame feature fusion, a deformable cross-frame Transformer module is proposed. This module incorporates independently learnable offset parameters for different frame features, enabling the model to adaptively correlate dynamic features across multiple frames and improve the temporal information utilization of the model. A proposal-aware sampling algorithm is introduced to significantly increase the foreground point recall, further optimizing the efficiency of feature extraction. The obtained multi-scale and multi-frame voxel features are subjected to an adaptive fusion weight extraction module, referred to as the proposed mixed voxel set extraction module. This module allows the model to adaptively obtain mixed features containing both spatial and temporal information. The effectiveness of the proposed algorithm is validated on the KITTI, nuScenes, and self-collected urban datasets. The proposed algorithm achieves an average precision improvement of 2.1% over the latest multi-frame-based algorithms.

**Keywords:** autonomous vehicle; 3D object detection; Transformer; point clouds





## 1. Introduction

In the continuous evolution of autonomous driving technology, environmental perception, as a critical domain, has attracted extensive academic research and engineering practices. One significant task within this domain is to accurately acquire 3D positional and classification information of surrounding objects for vehicles under complex weather conditions and road environments. Currently, numerous 3D detection algorithms employ deep learning methods to extract features from images [1–6], point clouds [7–12], and fusion data [13–20], achieving remarkable results on commonly used public datasets [21–23]. However, the data acquired during the vehicle's travel exhibits spatiotemporal continuity, and existing methods predominantly focus on single-frame data, neglecting to fully exploit the semantic correlations between historical information. While the motion of an object

changes over time, the environmental information and object features between adjacent frames remain similar. In traffic environments, perception algorithms are susceptible to various environmental interferences, while objects also exhibit feature discrepancies due to changes in perspective. These influences result in abrupt variations in the size and orientation of 3D object bounding boxes in continuous time sequences. Human perception benefits from the historical information of adjacent frames, aiding in obtaining more accurate predictions for the perception system.

LiDAR sensors offer rich spatial structural information, and mainstream 3D object detection algorithms typically adopt point cloud-based methods. Due to the unordered and non-structural nature of point cloud data, it is challenging to directly leverage feature extraction networks, as in the case of images [24–27], to obtain multiscale features. Existing approaches address this issue by voxelization [7–9,12,28,29] or Bird's Eye View (BEV) projection [30–32] of the raw point cloud, followed by utilizing 3D convolutional neural networks to extract various spatial features. While these methods tackle the difficulty in point cloud feature extraction, they struggle to handle feature deficiency when objects are distant or occluded. In recent research, some methods attempt to tackle 3D object detection challenges by leveraging information from multiple frames [33–39]. One easily introduced approach involves feeding features derived from sequences of multiple frames of point clouds into Long Short-Term Memory (LSTM), iteratively producing object detection results [35,39]. However, this method requires sequential data processing, making it less amenable to parallel computation of multiple frames.

Transformer [40], originally designed for natural language processing, achieves parallel context aggregation through attention mechanisms. Since its introduction into the field of object detection in recent years, Transformer has been widely adopted and demonstrates significant effectiveness. Therefore, some studies [33,36,38] explore the use of Transformer to associatively fuse multi-frame point cloud data, enhancing the spatial information of detected objects. However, in current multi-frame-based object detection methods, attention mechanisms are predominantly applied in the later stages. This is because the motion of objects leads to continuous but varying frame-wise mismatched features. The same object has different spatial features near and far, and different positions on the feature map after the feature extraction layer. Consequently, directly fusing data from multiple frames at the same position through concatenation becomes impractical. While the encoder of a Transformer can perform attention calculations for each feature vector between two frame feature maps, substantial attention computations are required in the early stage of feature extraction to learn spatial relationships between different frames. To address this challenge, we propose a 3D object detection algorithm based on a Spatiotemporal Deformable Transformer (SDT) for integrating features across multiple frames and scales. Inspired by deformable convolutional neural networks [41], DEtection TRansformer (DETR) [42], and deformable DETR [43], our research combines and extends their principles to the task of 3D feature fusion. The proposed approach first employs 3D submanifold sparse convolution [44] to obtain a set of multi-scale features. Subsequently, selective matching of Query and Key is achieved by setting multiple learnable offset parameters. By avoiding self-attention weight computations for all feature vectors, our method reduces computational requirements and accelerates model convergence speed.

In addition, the huge amount of raw point clouds significantly escalates the computational demands of the model, posing challenges in meeting real-time requirements. Several methods [7,11,45] employ the Farthest Point Sampling (FPS) to acquire a fixed number of key points. However, key points obtained through FPS-based algorithms often include numerous background points, introducing irrelevant background features into the model training process. Although this method can obtain point clouds covering the whole scene evenly, the number of point clouds contained in distant objects will be further reduced, which brings difficulties to the object detection task. Therefore, we propose a method named Proposal Aware Sampling (PAS) to enhance sampling efficiency. This approach leverages the Region Proposal Network (RPN) to eliminate background points from 3D

proposal boxes, thereby effectively obtaining a greater number of foreground points. By reducing background noise points, this method simultaneously avoids the loss of points belonging to the object.

In summary, this paper makes the following contributions:

(1) Addressing the issue of fine-grained feature misalignment in deformable attention mechanisms, we propose a Deformable Cross-Scale Transformer (DCST) module. This module employs a multi-scale offset mechanism, enabling non-uniform sampling of features across different scales. This enhances the spatial information aggregation capability of the output features.

(2) Tackling the problem of feature misalignment caused by object motion in multi-frame feature fusion, we introduce a Deformable Cross-Frame Transformer (DCFT) module. Independent learnable offset parameters are assigned to different frames, allowing the model to adaptively associate dynamic features across multiple frames, thereby improving the model's utilization of temporal information.

(3) To fully exploit information from cross-scale and cross-frame features, we present a Hybrid Voxel Set Extraction (HVSE) module. By predicting fusion weights for two types of feature vectors, the model adaptively obtains hybrid features containing both spatial and temporal information. We also devise a Proposal-Aware Sampling (PAS) algorithm to enhance foreground point recall rates, further optimizing the model's feature extraction efficiency.

## 2. Related Work

### 2.1. 3D Object Detection Based on Multi-Frame Data

The challenge in processing temporal information lies in the mismatch of the object's position across different frames relative to the ego-motion of the vehicle. This results in spatial inaccuracies in the fused features obtained by directly concatenating data from different frames. Consequently, numerous studies have explored various methods for appropriate feature extraction and encoding of temporal information. In work [35], the utilization of LSTM, known for handling sequential data, involves employing a 3D sparse convolutional U-Net module to independently process multi-frame point clouds for spatial feature encoding. This model integrates features from the current frame with the hidden and memory features from the previous frame, subsequently outputting updated hidden and memory features. The proposed network detection head, graph convolutional neural network, and maximum suppression algorithm are then applied to obtain the detection results. Similarly, in another work [39], PointNet encoding is employed to process each single-frame point cloud datum. Subsequently, a convolutional LSTM network is applied sequentially to extract temporal information from multiple frames, and the final detection results are obtained through existing detection heads.

However, LSTM-based methods only use a single memory state and bring a lot of calculations. In work [46], a comprehensive approach is presented, employing entire point cloud sequences as input. This approach adopts a modular design and a series of custom deep network models, including a multi-frame detector, tracker, and object auto-annotation model. Initially, the multi-frame detector is employed for the initial object localization. Subsequently, tracking is utilized to extract all relevant object point clouds and detection boxes, which are then forwarded to subsequent models for further processing. Ultimately, this pipeline can output finely processed object trajectories at the bounding box level. While this method achieves precise object detection, achieving real-time processing remains challenging. The work [32] compresses the height information of 3D point clouds in each single-frame to a 2D plane. Subsequently, a feature extraction network is employed to individually extract features from multi-frame Bird's Eye View (BEV) data. The features are then weightedly fused using 1D convolution, and a detector is applied to output detection boxes. Tracking is accomplished based on the positional relationships among these detection boxes. In the fusion stage, this method assigns different weights to features from different frames. The final stage involves multi-frame association

through a tracking algorithm. A similar approach is adopted in work [34], where point cloud data is first transformed into BEV pseudo-images. The features from different frames are then dimensionality-reduced and flattened as inputs to the encoder. These methods do not directly address the issue of feature misalignment during the fusion stage.

Therefore, some approaches utilize proposal network output, such as bounding boxes or centroid points, to guide the association of features across multiple frames, as bounding boxes can directly provide spatial positional information for different objects. The method [33] first utilizes PointPillars [9] to obtain a large number of bounding boxes, then mapping these boxes to multi-frame features to derive local features for each bounding box. Subsequently, a cross-attention mechanism is employed to achieve multi-perspective feature alignment and aggregation across frames, with these features being preserved and continuously updated. In the final stage, a 3D detection head is employed to output the position and classification information of the object. Another similar approach [36] involves the proposed spatiotemporal fusion module, which performs weighted fusion on BEV features of adjacent frames, followed by predicting the central position of the objects to guide the extraction of multi-frame fusion features. However, these methods do not fully leverage the early features with rich semantic information. Hence, the multi-frame fusion module proposed in this paper associates information across frames at earlier stages.

### 2.2. 3D Object Detection Based on Attention Mechanism

DETR [42] first introduced the thought of Transformers [40] into image object detection. Initially, it unfolds extracted image features into multiple sets of feature vectors, incorporating positional encoding information as queries in self-attention computations. Subsequently, following the encoder-decoder framework, a fixed number of bounding box results are obtained. Leveraging the computed self-attention results, the Transformer encoder facilitates adaptive weighted fusion of local and global features. Consequently, numerous approaches in spatial feature fusion have taken advantage of this benefit in 3D object detection based on point clouds.

The work [10] initially employs FPS for downsampling the point cloud and generates local regions through a spherical query module. Subsequently, it utilizes Transformer blocks, treating point features and coordinates as inputs, to generate aggregated features for local regions. To further refine the centroid points, an attention map from the last Transformer layer is applied to fine-tune the coordinates of the object bounding box. The method [47] adopts a series of Transformer-based stacked sub-modules, where each Transformer-based sub-module comprises a self-attention layer to enhance contextual relationships and uniqueness for each voting cluster feature. Additionally, it includes a cross-attention layer for positioning alignment with the initial voting cluster, ensuring that the refined voting clusters remain consistent with the input point cloud. Several methods [48,49] attempt to directly implement spatial attention mechanisms based on point features. They achieve this by initially predicting points relevant to the object through processing the raw point cloud. Subsequently, each sampled point feature vector is utilized as a query input to the encoder. To further reduce computational demands, the work [50] calculates self-attention on point features within proposed bounding boxes after obtaining a series of proposals. Following this, a fully connected layer refines the decoder's output features of the bounding boxes.

Unlike image-based attention mechanisms, the spatial feature map derived from voxelization of point clouds is sparse. Applying self-attention computation to each voxel feature, as done in methods similar to DETR, would result in significant computational overhead and inefficiency. Therefore, the proposed approach in this paper introduces deformable convolutions into the computation process of the Transformer encoder, extending it to 3D point cloud detection. By incorporating a set of learnable offsets, the method ensures that each query only needs to compute attention weights with the most relevant multi-scale voxel features. This not only reduces computational complexity, but also allows the model to adapt to objects of various shapes.

## 3. The Framework of the Deformable Spatiotemporal Transformer

In addressing the aforementioned challenges, this paper proposes a 3D object detection network based on a deformable spatiotemporal Transformer. As illustrated in Figure 1, the algorithm takes multiple frames of point clouds as input and effectively integrates features from multi-scale, multi-frame point clouds into key points using a point-voxel [7] feature aggregation approach. Initially, to facilitate the processing of point clouds, a voxelization [28] method is employed to partition the point cloud into regular cubic structures. Subsequently, 3D sparse convolution [7,8] modules are applied to the adjacent N frames of point clouds, extracting voxel spatial features for each frame individually. The Deformable Cross-Scale Transformer (DCST) module is proposed to selectively fuse features from multiple scales within the current frame, resulting in intra-frame fusion features. This facilitates the exchange of information between deep and shallow features. Simultaneously, the Deformable Cross-Frame Transformer (DCFT) module applies similar operations, fusing same-scale features from the past N-1 frames into the current frame, thereby obtaining inter-frame fusion features. This enables the exchange of object features at multiple positions and angles. Multiple-Scale Offset (MSO) and Multi-Frame Offset (MFO) are employed to explore the correlation between the Query and reference points at various positions. The hybrid voxel set abstraction module introduces a learnable weight parameter to further integrate intra-frame and inter-frame fusion features, generating the ultimate mixed voxel features. Following voxelization and 3D sparse convolution, there is an inevitable loss of high fine-grained point cloud spatial features.

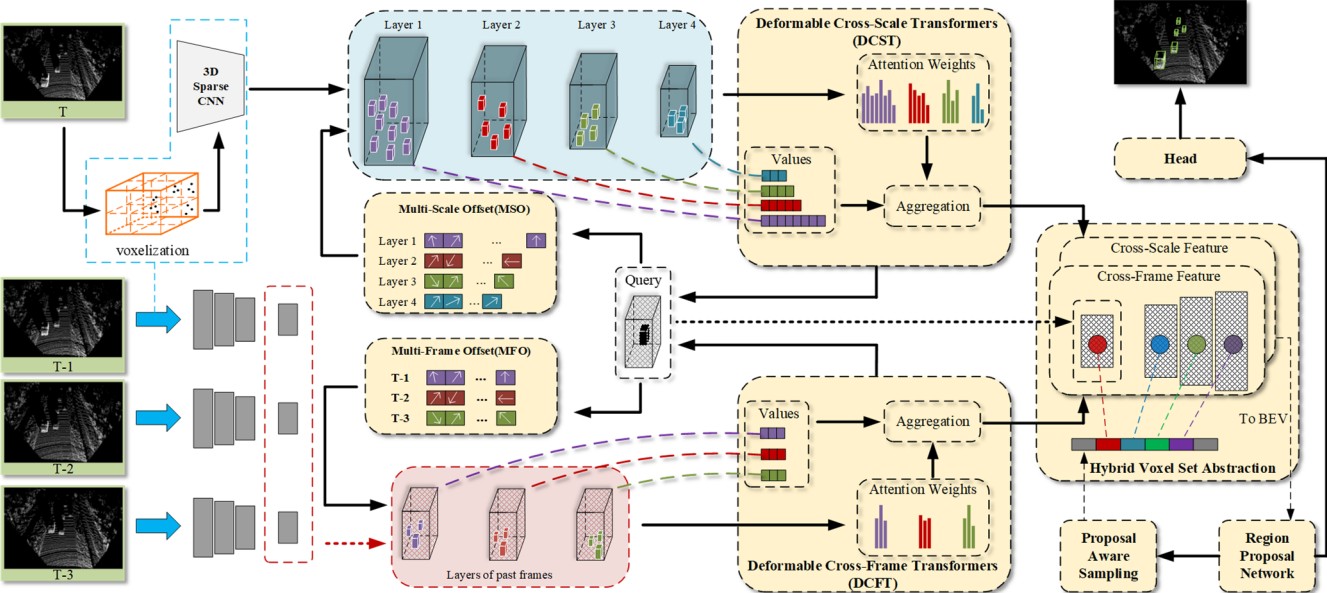

**Figure 1.** The framework of the deformable spatiotemporal transformer.

Therefore, the proposed method employs key points [7] to obtain point-based local voxel features from hybrid voxel features using PointNet++ [51]. Simultaneously, to enable the point cloud downsampling algorithm to capture more foreground points, a proposal-aware sampling algorithm is proposed. Ultimately, all key points' features are fused with reference points within the proposal, and after further refinement by the detection head, the 3D position and classification of the object are obtained.

### 3.1. 3D Feature Extraction

3D sparse convolution is a widely used method for feature extraction from point clouds, particularly suited for handling sparse data. 3D sparse convolution effectively reduces computational load and storage requirements by performing convolution operations only in the non-empty regions of the data. This is achieved by using a sparse tensor, which stores

only the non-zero data points and their location information within the dataset. Assuming there is an input feature map or image *X* and a convolution kernel *K*, the traditional 3D convolution operation can be expressed as:

$$Y(i,\ j,\ k) = \sum_{p=-a}^{a} \sum_{q=-b}^{b} \sum_{r=-c}^{c} K(p,q,r) \cdot X(i+p,\ j+q,\ k+r) \tag{1}$$

where $(i,\ j,\ k)$ refers to a certain location of the output feature map $Y$, $(p,q,r)$ refers to the relative position of the convolution kernel *K*, and $(a,b,\ c)$ is the size of convolution kernel *K*. For 3D sparse convolution, the computation is performed only at non-zero positions. Therefore, the operation of 3D sparse convolution can be formalized as:

$$Y(i,\ j,\ k) = \sum_{(p,q,r)\in\mathrm{N}(i,\ j,\ k)} K(p,q,r) \cdot X(i+p,\ j+q,\ k+r) \tag{2}$$

where $\mathrm{N}(i,\ j,\ k)$ refers to a set of nonzero feature points at location $(i,\ j,\ k)$ of the input feature map $Y$. Like the settings of current works [7,44,52], the input point cloud is voxelized to L × H × W, which would be down-sampled to voxel features through 4-scale $3 \times 3 \times 3$ 3D sparse convolution with $1\times, 2\times, 4\times, 8\times$ size.

### 3.2. Deformable Cross-Scale Transformer Module

Features at different scales can capture various levels of fine-grained information. Deep features, owing to their enlarged receptive fields, prioritize global contextual information. Conversely, shallow features focus more on local information. Therefore, Feature Pyramid Networks (FPN) are frequently incorporated after the feature extraction layers to aggregate features at multiple scales. The concept of deformable DETR [43] integrates the idea with the approach proposed in DETR [42], utilizing learnable offset parameters to guide information exchange between different layers, thereby replacing the role of FPN. Our work proposes a Deformable Cross-Scale Transformer (DCST) that extends this methodology to 3D space and enhances its performance. The proposed DCST module is illustrated in Figure 2. This module enhances the deformable DETR by introducing multiple sets of varying offsets for each reference point, corresponding to multiple layers of spatial voxel features.

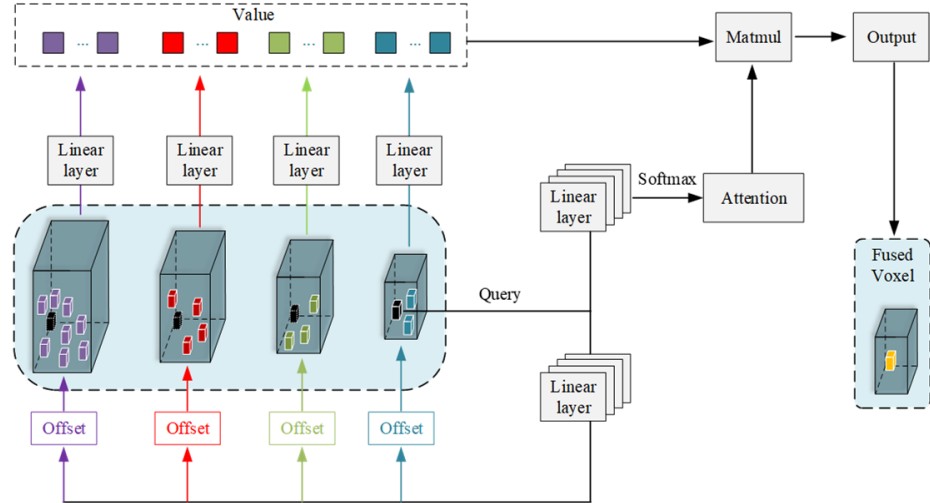

**Figure 2.** The deformable cross-scale Transformer module.

In the original approach, each reference point generates only one set of identical offsets for each layer of feature maps. Subsequently, the reference point and sampling point positions are mapped to different layers of feature maps through the downsampling rate.

However, different layers contain features with distinct receptive fields: shallow layers embody semantic details, while deep layers embody global features. Consequently, due to variations in receptive fields, shallow features need an increased number of sampling points for object description. The proposed method DCST leverages the features of reference points to generate Query vectors. These vectors are then projected through multiple linear layers to predict offset parameters for different quantities and positions of features at each layer's voxel. Each sampling point feature is transformed into a Value vector, and its corresponding attention weight is directly predicted by the Query vector through multiple linear layers. Assuming the Query vector, denoted as $\mathbf{q}_{mli}$, for the $i$-th sampling point in the $l$-th layer within the $m$-th attention head, the predicted attention weight $A_{mli}$ can be represented as follows:

$$A_{mli} = \text{Softmax}\big(L_1(\mathbf{q}_{mli}),\ L_2(\mathbf{q}_{mli}),\ \ldots, L_j(\mathbf{q}_{mli})\big) \tag{3}$$

where $L_j$ denotes the linear layer, $A_{mli}$ representing the degree of correlation between the reference point and each sampled point. The sampled points are obtained by adding an offset $\mathbf{\Delta p}^{\mathrm{q}}_{mli}$ to the coordinates of the reference point center $\mathbf{p}^{\mathrm{q}}_{mli}$. Unlike deformable DETR, the coordinates corresponding to the sampling points and offsets are in three-dimensional space. $\mathbf{\Delta p}^{\mathrm{q}}_{mli}$ can be represented as follows:

$$\mathbf{\Delta p}^{\mathrm{q}}_{mli} = L_1(\mathbf{q}_{mli}),\ L_2(\mathbf{q}_{mli}),\ \ldots, L_j(\mathbf{q}_{mli}) \tag{4}$$

Finally, the obtained fused feature vector can be represented as:

$$\mathbf{f}^s_{li} = \sum_{m=1}^{M} W^s_m \Big[ \sum_{l=1}^{L} \sum_{i=1}^{I} A_{mli} \cdot W^v_{mli} \cdot G(\phi(\mathbf{p}^{\mathrm{q}}_{mli}) + \mathbf{\Delta p}^{\mathrm{q}}_{mli}) \Big] \tag{5}$$

where $\phi(\cdot)$ refers to the normalization function, employed for normalizing coordinates to the [0, 1] range, facilitating the mapping of reference points to distinct voxel features in different layers. The matrix $W^v_{mli}$ represents the projection of sampled point features onto the Value vector, $W^s_m$ denotes the learnable multi-head attention weights in cross-scale fusion, and $G(\cdot)$ denotes the spatial trilinear interpolation function. The introduction of this function is necessitated by the fact that the learned values of $\mathbf{\Delta p}^{\mathrm{q}}_{mli}$ are mainly fractional, making it challenging to precisely align with voxel coordinates. As shown in Figure 3, assuming that there is a reference point $f(x_d, y_d, z_d)$ with eight0 voxel center points nearby, the spatial trilinear interpolation can be calculated as follows:

$$f_{Ci} = f_{Li}(1 - x_d) + f_{Ri}x_d, i = 1, 2, 3, 4 \tag{6}$$

$$\begin{cases} f_{V1} = f_{C2}(1 - y_d) + f_{C1}y_d \\ f_{V2} = f_{C3}(1 - y_d) + f_{C4}y_d \end{cases} \tag{7}$$

$$f = f_{V1}(1 - z_d) + f_{V2}z_d \tag{8}$$

Through the integration of the DCST module, fused voxel features are obtained, and the ultimate intra-frame fused feature can be expressed as:

$$F_{\text{intra}} = \{F^s_1, F^s_2, F^s_3, F^s_4\} \tag{9}$$

where $F^s_1, F^s_2, F^s_3, F^s_4$ refer to four scales of voxel features after fusion.

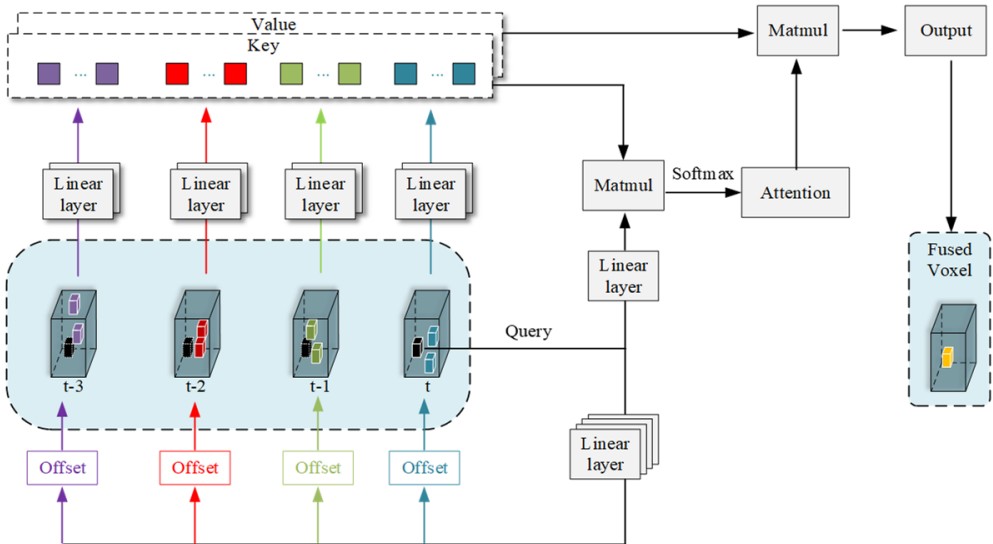

**Figure 3.** The deformable cross-frame Transformer module.

### 3.3. Deformable Cross-Frame Transformer Module

There are temporal contextual connections between different frame data, and the object features are similar in multi-frame data. Commonly used cross-attention mechanisms can obtain correlation results between two inputs, and learnable fusion weights can be derived based on this correlation. However, directly applying attention mechanisms to multi-frame data introduces a substantial computational complexity. Therefore, we propose a Deformable Cross-Frame Transformer (DCFT) that integrates each layer of multiscale voxel features with adjacent frames. The fusion of inter-frame information differs from the intra-frame multiscale fusion, as the object's movement with respect to the ego car leads to voxel features distribution at different positions. However, in the original Deformable DETR, attention weights are directly obtained from the Query vector. This results in challenges during training to determine the correct offset for position and appropriate weight parameters. To address this issue, as shown in Figure 3, DCFT utilizes sampled point features to generate Query and Key vectors separately, applying attention computation to both. The Value vectors generated from reference points are fused to the original Query position based on attention weights. The computation of attention weight matrix for the *i*-th sample point in the *f*-th frame within the *m*-th attention head can be expressed as:

$$A_{mfi} = \mathrm{Softmax}\left(\frac{\left(W_{mf}^q q_{mfi}\right)^T \left(W_{mf}^k k_{mf1}\right)}{\sqrt{d}},\right.$$
$$\left.\frac{\left(W_{mf}^q q_{mfi}\right)^T \left(W_{mf}^k k_{mf2}\right)}{\sqrt{d}}, ..., \frac{\left(W_{mf}^q q_{mfi}\right)^T \left(W_{mf}^k k_{mfj}\right)}{\sqrt{d}}\right) \tag{10}$$

where $W_{mf}^q$ and $W_{mf}^k$ refer to the projection matrix of Query and Value, and $\sqrt{d}$ is the scale factor used to prevent excessive scaling of the dot product sum. Similar to the proposed DCST, the calculation of reference point features for cross-frame fusion can be expressed as:

$$\mathbf{f}_{fi}^f = \sum_{m=1}^{M} W_m^f \left[\sum_{f=1}^{F} \sum_{i=1}^{I} A_{mfi} \cdot W_{mfi}^v \cdot G(\phi(\mathbf{p}_{mfi}^q) + \Delta \mathbf{p}_{mfi}^q)\right] \tag{11}$$

where $W_{mfi}^v$ refers to the projection matrix that projects sampled points to the Value vector, and $W_m^f$ denotes the learnable multi-head attention weights in cross-frame fusion. It should be noted that the number of learnable offset values for each frame in DCFT is fixed. This is due to the fact that the fusion in cross-frame data involves voxel features from the same layer, sharing identical receptive field sizes and containing object features at the same fine-grained level. The objective of multi-frame fusion is to establish associations between voxel

features of the current frame and voxel features from adjacent frames at different positions but representing the same object. Through the fusion process in the DCFT module, fused voxel features are obtained, and the final inter-frame fusion feature can be represented as:

$$F_{\text{inter}} = \left\{ F_1^f, F_2^f, F_3^f, F_4^f \right\} \tag{12}$$

where $F_1^f, F_2^f, F_3^f, F_4^f$ refer to the voxel features of the adjacent four frames after fusion.

### 3.4. Proposal-Aware Sampling Module

Following the processing of voxel features by the proposed DCST and DCFT, information from multiscale and multi-frame point clouds is, respectively, integrated into the voxel features at the corresponding positions of the current frame. While voxel-based methods efficiently extract features, they inevitably lead to the loss of spatial details. To address this issue, a point-voxel based method of aggregating 3D voxel features through key points is applied in the subsequent stages.

Common FPS methods can uniformly sample the original point cloud but may include a substantial number of background points. Therefore, a Proposal-Aware Sampling (PAS) algorithm is proposed to enhance the recall rate of foreground point sampling for FPS. Voxel features within a single frame and across multiple frames are compressed into BEV features. Subsequently, a Region Proposal Network (RPN) is employed to generate proposal boxes for foreground points. Assume N proposal boxes $B_j \in \left[ C_j, d_j, h_j, w_j \right]$, where $C_j, d_j, h_j, w_j$ represent the center point, depth, height, and width of the proposal box, respectively. The Euclidean distance between each point $P_{Li}$ in the original point cloud and $C_j$ is calculated to filter out background points, expressed as:

$$\begin{aligned}
D_{ij} &= \left\| P_{Li} - C_j \right\|, \\
P_{Li} &= \left[ x_i^L, y_i^L, z_i^L \right], \\
C_j &= \left[ x_j^P, y_j^P, z_j^P \right], \\
D_{ij} &= \left[ \Delta x_{ij}, \Delta y_{ij}, \Delta z_{ij} \right],
\end{aligned} \tag{13}$$

so that if the $D_{ij}$ is greater than $\left[ d_j/2, h_j/2, w_j/2 \right]$, it indicates that the point is a background point and should be deleted.

As shown in Figure 4, visual results obtained through different sampling algorithms for the target vehicle indicate that the proposed PAS approach effectively filters out a greater number of background points. It is evident that the method based on FPS loses significant details of the object point cloud and exhibits the highest proportion of background points. The Sectorized Proposal-Centric (SPC) method [53] directly employs the longest edge of the proposal box as the diameter to form a spherical region as the criterion for selecting foreground points. However, this method tends to include excessive background points. The proposed PAS employs filtering criteria that are more aligned with the actual shape of the object, thereby obtaining points more relevant to the object.

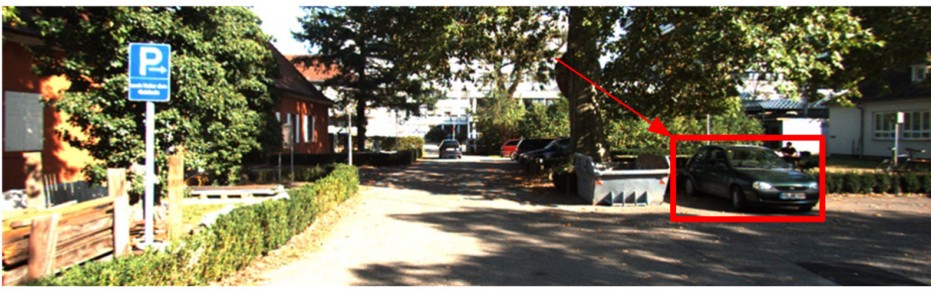

(**a**)

**Figure 4.** *Cont.*

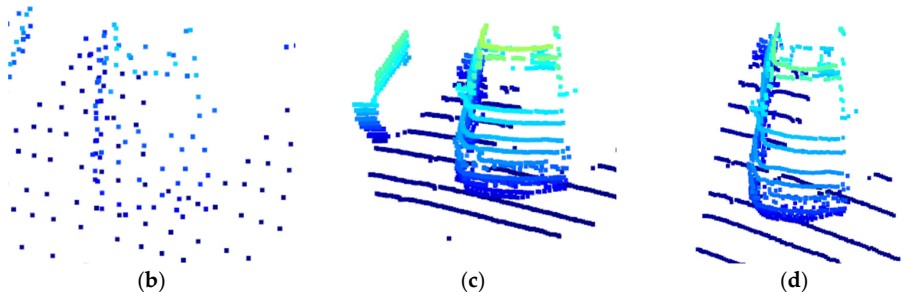

**Figure 4.** The visualization results of different sampling methods. (**a**) The target vehicle in the original image; (**b**) FPS; (**c**) SPC (**d**) PAS (proposed).

*3.5. Hybrid Voxel Set Extraction Module*

Based on the key points obtained from PAS, a Hybrid Voxel Set Abstraction (HVSA) is proposed, which can aggregate local voxel features in cross-scale voxel features and cross-frame voxel features by voxel set abstraction [7]. Specifically, assume that $F_{lP} = \{f_{l1}^c, ..., f_{lP}^c\}$, $c = [\text{inter, intra}]$ denotes the i-th layer feature of voxel features and $V_{lP} = \{v_{l1}^c, ..., v_{lP}^c\}$ denotes 3D coordinates corresponding to each voxel feature. For the key point $P_{Kj}$, the allocation of adjacent voxel features within the radius r is obtained as follows:

$$S_{lj}^c = \left\{ [f_{lp}^c; v_{lp}^c - p_{Kj}]^T, \quad \begin{array}{c} \left\| v_{lp}^c - p_{Kj} \right\|^2 < r, \\ \forall f_{lp}^c \in F_{lp}, \\ \forall v_{lp}^c \in V_{lp} \end{array} \right\}$$ (14)

where $v_{lp}^c - p_{Kj}$ refers to local coordinates, however, due to the abstraction of varying amounts of voxel features by different key points, it is necessary to uniformly encode the neighboring voxels obtained for each point. This process can be expressed as:

$$F_{lj}^c = max\left\{ G\left( M\left( S_{lj}^c \right) \right) \right\}$$ (15)

where $M(\cdot)$ denotes random sampling of a certain number of voxel features from adjacent voxel features, and $G(\cdot)$ denotes a multi-layer fully connected network for encoding local voxel features and local coordinate features. Subsequently, all voxel features within the range are integrated into a key point through maximum channel pooling. In the end, the multi-scale and multi-frame voxel features for each key point at each layer are concatenated together as follows:

$$F_j^c = \left[ F_{1j}^c, F_{2j}^c, F_{3j}^c, F_{4j}^c \right]$$ (16)

As shown in Figure 5, Equations (12)–(14) extract multi-scale and multi-frame information from the intra-frame voxel features and inter-frame voxel features, obtaining $F_j^{\text{intra}}$ and $F_j^{\text{inter}}$ at the key points. To further integrate the two types of information, a weight prediction module is proposed, enabling weighted fusion based on learnable weights. It is worth noting that, due to the attention mechanisms implemented by DCST and SCFT, different-layer feature information exchange for a single-frame and same-layer information exchange for multi-frame have been achieved. Moreover, these features have already been aggregated at the same key point position. Therefore, attention calculations between spatial positions are not required for this fusion stage. There will be a learning weight vector for each key point. The calculation can be expressed as:

$$w_j^{keypoint} = \sigma[G(\tan h(G(F_j^{\text{intra}}) + G(F_j^{\text{inter}})))]$$ (17)

where $G(\cdot)$ denotes a multi-layer fully connected network, $\tanh(\cdot)$ denotes hyperbolic tangent activation function, and $\sigma$ refers to the sigmoid function used to normalize $w_j^{keypoint}$ into the range [0, 1]. Finally, the obtained features of each key point can be expressed as:

$$F_j^{keypoint} = F_j^{\text{intra}} w_j^{keypoint} \oplus F_j^{\text{inter}} \left(1 - w_j^{keypoint}\right) \qquad (18)$$

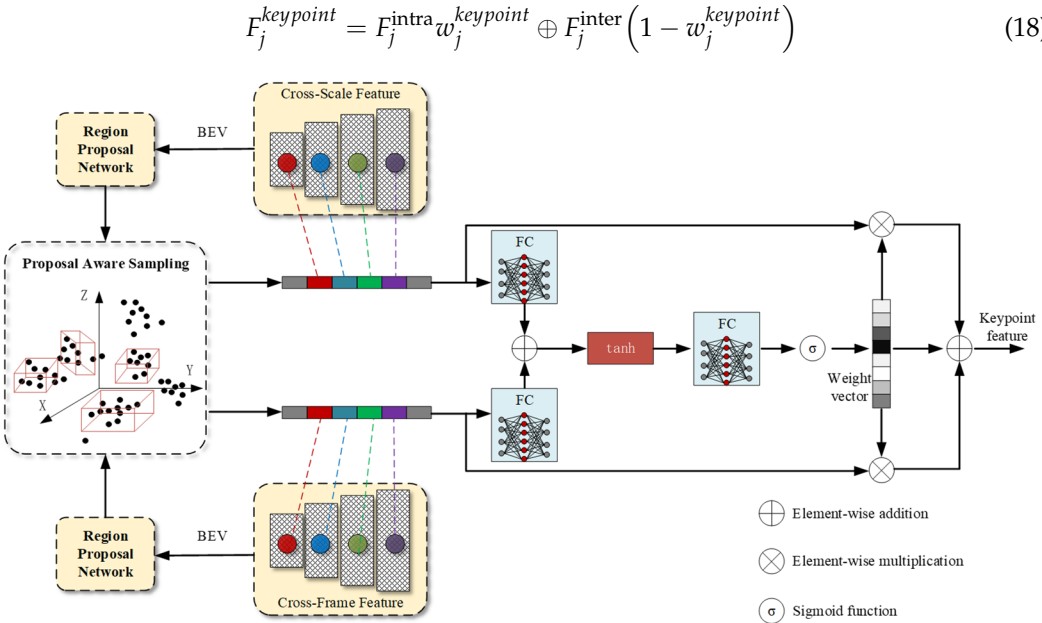

**Figure 5.** The diagram of the hybrid voxel set extraction module.

## 4. Experiments

### 4.1. Experiment Setup

#### 4.1.1. Voxelization

The proposed algorithm was validated on the nuScenes dataset and the KITTI dataset. The raw point cloud was voxelized into regular voxels before serving as input. For the nuScenes dataset, the point cloud range was clipped to: X-axis and Y-axis [−75.2, 75.2] m, Z-axis [−2, 4] m. The input voxel size was set to [0.1 m, 0.1 m, 0.15 m]. Since the KITTI dataset provides annotations only for objects within the field of view (FOV), we clipped the point cloud range to: X-axis [0, 70.4] m, Y-axis [−40, 40] m, and Z-axis [−3, 1] m. The input voxel size was set to [0.05 m, 0.05 m, 0.1 m].

#### 4.1.2. Public Dataset and Training Setup

(a)  nuScenes dataset

The nuScenes dataset [22] comprises 700 sequences for training and 150 sequences for validation. Each point cloud sequence has a duration of approximately 20 s, with a frame interval of 0.05 s. Annotations are provided for every ten consecutive frames, referred to as keyframes. The nuScenes dataset provides a substantial volume of data and covers a diverse range of environmental conditions, including variations in times of day and weather. It offers detailed annotations across a variety of object classes and sensor modalities, supporting an extensive evaluation of the model's capabilities in multi-modal 3D detection scenarios. Additionally, the inclusion of continuous multi-frame data in the nuScenes dataset enables the application of spatiotemporal fusion methods. The presence of sequential frame data enables the evaluation of the model's ability to utilize temporal information, which is crucial for accurate 3D object detection in dynamic environments typical of autonomous driving.

The primary evaluation metric for the detection task is the mean average precision (mAP). The mAP computation utilizes a series of centroid distance thresholds rather than the commonly used bounding box IoU thresholds. Training is conducted using data from

keyframes taken every four consecutive frames due to the absence of ground truth for non-keyframes.

(b) KITTI dataset

The KITTI dataset [21] consists of 7481 training samples and 7518 testing samples, with the training samples typically divided into training subsets (3712 samples) and validation subsets (3769 samples). This dataset is one of the most established benchmarks in the field of autonomous driving. It provides a diverse set of real-world scenarios, which are crucial for evaluating the performance of our 3D sparse convolution models in realistic settings. The KITTI dataset includes a variety of annotated 3D objects, such as vehicles and pedestrians, captured in different urban environments, making it highly suitable for testing the accuracy and robustness of our model. Since KITTI does not provide consecutive frame data, the proposed model is equipped with only the fusion module of DCFT during testing.

(c) Training setup

For model training, An NVIDIA GTX 3090 graphics card (Nvidia Corporation, Santa Clara, CA, USA) was utilized for training with a batch size of four, across a total of 80 epochs for KITTI and 50 epochs for nuScenes, and the initial learning rate was set at 0.01. The operating system was Ubuntu 20.04, and the algorithm was implemented using the OpenPCDet ( version 0.5) and PyTorch ( version 1.10) frameworks, with Adam being employed as the optimizer.

### 4.1.3. Self-Collected Dataset

As shown in Figure 6, to further validate the portability and effectiveness of the proposed algorithm, we established a physical testing platform equipped with an 80-line LiDAR (RoboSense RS-Ruby Lite, 10 Hz, Shenzhen, China), an IMU (200 Hz), and a stereo camera (30 Hz). The LiDAR has a 360-degree horizontal field of view and a 40-degree vertical field of view. The camera has a resolution of 1080 × 720, a horizontal field of view of 71 degrees, and uses a global shutter to output images.

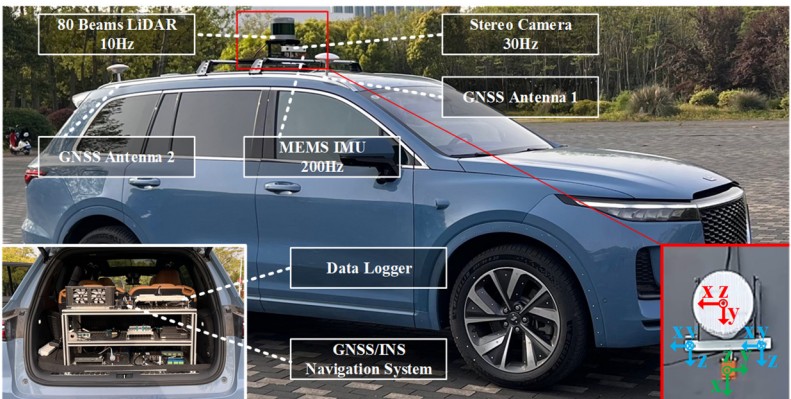

**Figure 6.** The hardware platform for the self-collected dataset.

We collected small-scale datasets across two different scenes totaling ten continuous segments (each approximately 5 min in length). The collection includes eight segments from urban residential areas and two from city streets in Anting Town, Shanghai, China. The dataset encompasses various object types including cars, pedestrians, bicycles, and motorcycles. Data from all sensors were gathered through ROS and synchronized to a common timestamp. The proposed algorithm was tested on LiDAR data, which were transformed into the format of the KITTI dataset.

### 4.2. Results of the nuScenes Dataset

The nuScenes dataset provides continuous frame data for 150 scenes, serving as the evaluation benchmark to validate the effectiveness of the proposed algorithm. As



shown in Table 1, the proposed DSTrans exhibits significant advantages compared to recent state-of-the-art methods. It achieves the highest detection accuracy in categories such as "car", "pedestrian", "truck", "motorcycle", "construction vehicle", "bicycle", and "average precision (mAP)". Competitive results are also obtained in other categories. In comparison to the latest multi-frame 3D object detection algorithm, TCTR [34], our method shows a performance improvement of 2.1%.

**Table 1.** Comparison of 3D object detection results for different methods on the nuScenes validation dataset (%). The best results are marked in bold. Abbreviations: construction vehicle (CV), pedestrian (Ped.), motorcycle (Motor.), and traffic cone (TC).

| Method | Data | Car | Ped. | Bus | Barrier | TC | Truck | Trailer | Motor. | CV | Bicycle | mAP |
|---|---|---|---|---|---|---|---|---|---|---|---|---|
| SARPNET [54] | | 59.9 | 69.4 | 19.4 | 38.3 | 44.6 | 18.7 | 18.0 | 29.8 | 11.6 | 14.2 | 32.4 |
| InfoFocus [55] | | 77.9 | 63.4 | 44.8 | 47.8 | 46.5 | 31.4 | 37.3 | 29.0 | 10.7 | 6.1 | 39.5 |
| MAIR [56] | Single- | 47.8 | 37.0 | 18.8 | 51.1 | 48.7 | 22.0 | 17.6 | 29.0 | 7.4 | 24.5 | 30.4 |
| PointPillars [9] | frame | 68.4 | 59.7 | 28.2 | 38.9 | 30.8 | 23.0 | 23.4 | 27.4 | 4.1 | 1.1 | 30.5 |
| PointPainting [57] | | 77.9 | 73.3 | 36.1 | **60.2** | **62.4** | 35.8 | 37.3 | 41.5 | 15.8 | 24.1 | 46.4 |
| WYSIWYG [58] | | 79.1 | 65.0 | 46.6 | 34.7 | 28.8 | 30.4 | 40.1 | 18.2 | 7.1 | 0.1 | 35.0 |
| 3DVID [37] | | 79.7 | 76.5 | 47.1 | 48.8 | 58.8 | 33.6 | **43.0** | 40.7 | 18.1 | 7.9 | 45.4 |
| TCTR [34] | Multi- | 83.2 | 74.9 | **63.7** | 53.8 | 52.5 | 51.5 | 33.0 | 54.0 | 15.6 | 22.6 | 50.5 |
| TransPillars [59] | frame | 84.0 | 77.9 | 62.0 | 55.1 | 55.4 | 52.4 | 34.3 | 55.2 | 18.9 | 27.6 | 52.3 |
| DSTrans (Ours) | | **86.2** | **78.3** | 60.9 | 58.2 | 54.9 | **54.2** | 35.1 | **57.4** | **19.8** | **28.7** | **53.4** |

As shown in Figure 7, compared to the commonly used PointPillars method, the proposed approach can output stable detection results in continuous multi-frame point cloud data. Figure 7 illustrates the detection results for pedestrians and vehicles near the road in three consecutive frames, comparing the two methods. When the pedestrian is at a distance, PointPillars fails to detect the pedestrian, and due to occlusion, the insufficient return point cloud results in the failure to correctly detect the three vehicles near the road. In contrast, the proposed DSTrans leverages data from adjacent frames, enhancing the feature representation of objects and producing accurate detection results.

Different voxelization settings lead to voxel grids with varying spatial resolutions, directly impacting the fine-grained features obtained by 3D sparse convolution. Therefore, various voxelization parameters were set, and a comparison of 3D detection performance on the nuScenes dataset was conducted. As shown in Figure 8, each voxel size was set to [0.05, 0.05, 0.2], [0.1, 0.1, 0.2], [0.2, 0.2, 0.2], and [0.4, 0.4, 0.2]. The results indicate that as the voxel size increases, the extracted features inevitably lose details, leading to a decrease in detection accuracy. Conversely, overly small voxels result in increased computational demands, reducing the number of point clouds contained in each voxel and consequently decreasing the expressive power of local features.

In rainy conditions, the quality of LiDAR point clouds is noticeably affected due to several reasons: (1) The presence of raindrops leads to scattering and absorption of the laser beam in the atmosphere, resulting in a reduction of beam intensity. This may cause a weakening of the received signal by the LiDAR, lowering sensitivity and detectable range. (2) After intersecting with raindrops, the laser beam may undergo multiple reflections, causing a single laser pulse to generate multiple return points in space. This may introduce redundant points in the point cloud, interfering with the accuracy of object shape and position. (3) Scattering effects of raindrops can blur the boundaries between the laser beam and objects on the ground. This blurring effect may result in less distinct object contours in the point cloud, thereby reducing spatial resolution. Figure 9 illustrates the detection results of the proposed method in rainy conditions. It is evident that the LiDAR point clouds associated with each object have significantly decreased, especially for distant objects. However, due to the incorporation of information across multiple frames, the network captures temporal correlations between consecutive frames. The proposed algorithm still detects the majority of objects under these challenging weather conditions.

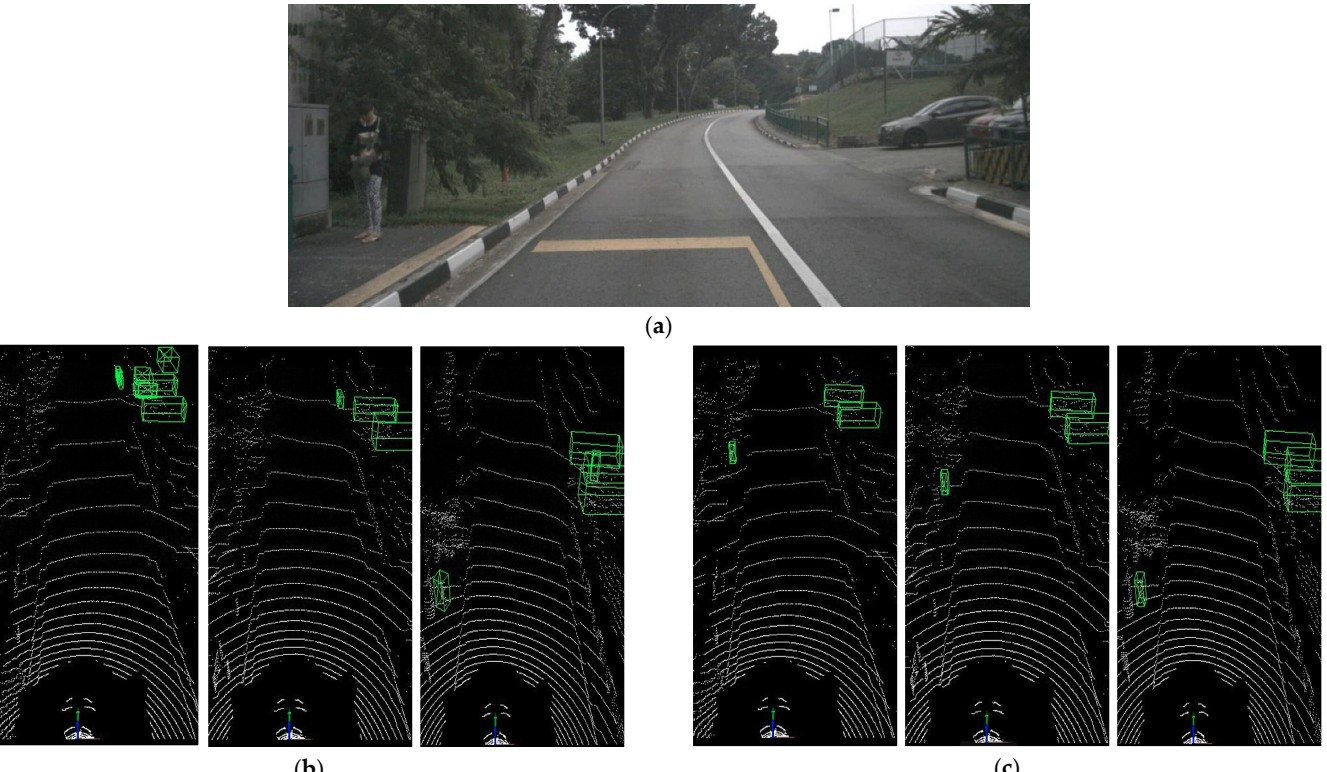

**Figure 7.** Visualization results of multi-frame point clouds in the nuScenes dataset. (**a**) The image of Frame *t*; (**b**) The results of PointPillars of three consecutive frames; (**c**) The results of proposed method of three consecutive frames. The green boxes represent the predicted boxes by algorithm.

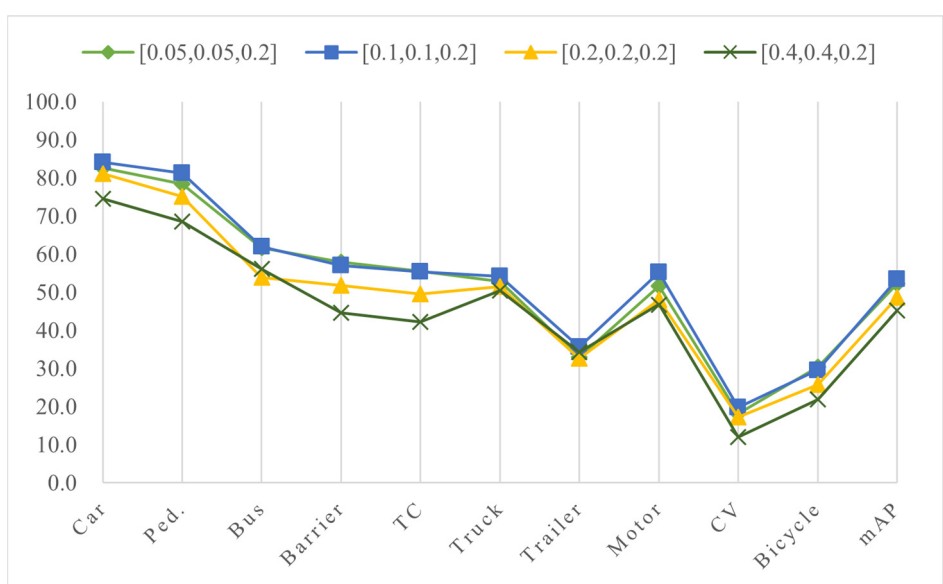

**Figure 8.** Comparison of detection results of different voxel setting parameters (%).

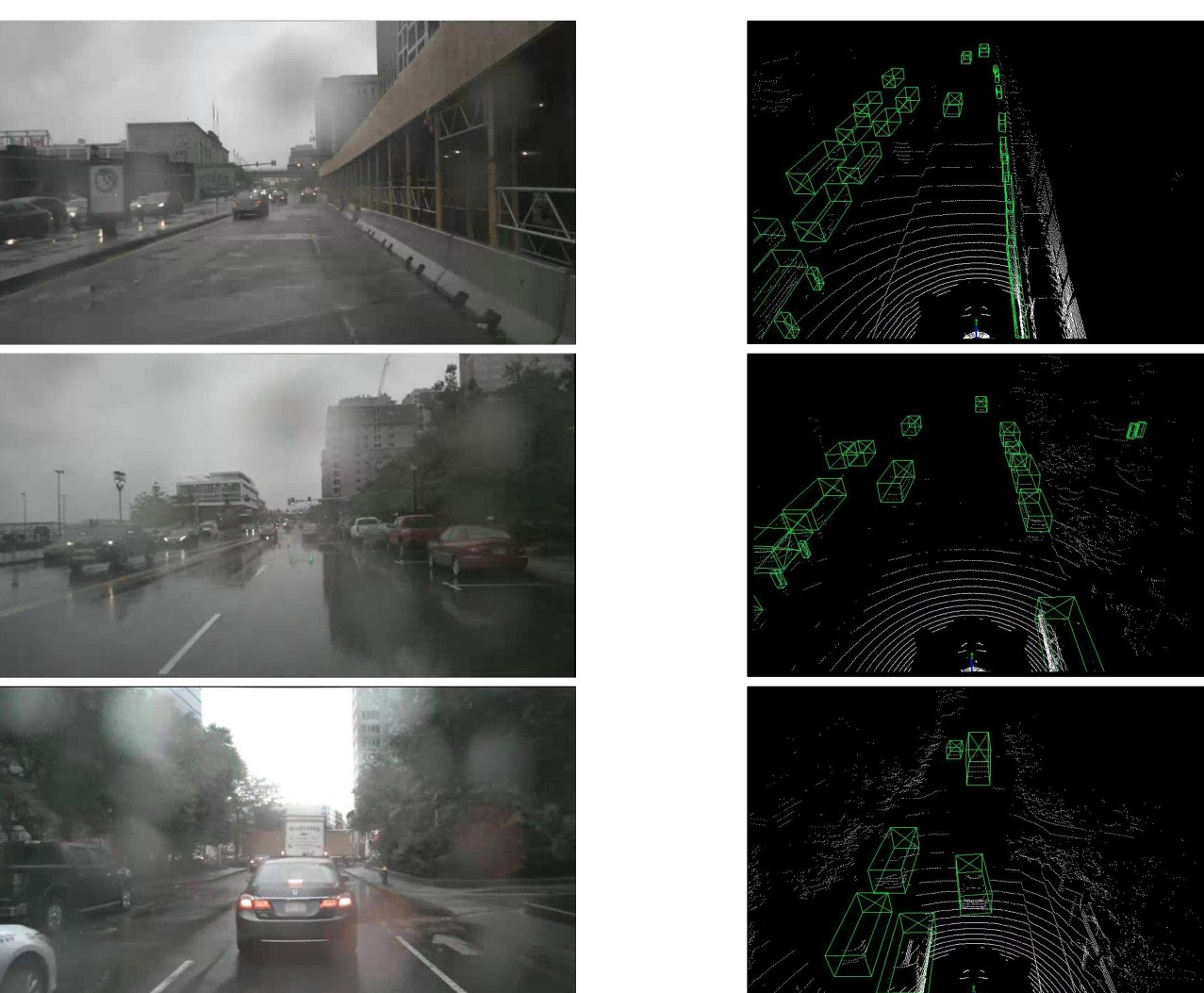

**Figure 9.** Detection results of the proposed method in rainy conditions. The green boxes represent the predicted boxes by algorithm.

*4.3. Results of the KITTI Dataset*

The test results of the KITTI dataset are commonly used 3D object detection evaluation metrics. Due to the absence of continuous temporal data in the 3D object detection task of the KITTI dataset, the proposed method in this paper excludes the multi-frame information fusion module DCFT, retaining only the DCST and PAS modules. As shown in Table 2, even without the inclusion of the DCFT module, the proposed method still achieves the highest mAP results. The architecture of PV-RCNN is most similar to the proposed algorithm, and by adopting the improvements presented in this chapter, a 4% improvement is observed in mAP results. The detection results of the "car", "pedestrian", and "bicycle" were increased by 1.4%, 8.3%, and 3.7%, respectively. All compared methods results are sourced from the official OpenPCDet data, and the proposed DSTrans is also implemented based on OpenPCDet. The proposed method achieves a single-frame processing speed of 0.18 s for the KITTI dataset on a GTX 3090 graphics card. Compared with PV-RCNN, the proposed model increases the number of learnable parameters by 16.1%, but the running time is only increased by 0.02 s.

**Table 2.** Comparison of 3D object detection results for different methods on the KITTI validation dataset (%). The best results are marked in bold. Abbreviations: pedestrian (Ped.), parameters (Params.).

| Method | Car | Ped. | Bicycle | mAP | Params. | Runtime |
|---|---|---|---|---|---|---|
| CaDDN [60] | 21.4 | 13.0 | 9.8 | 14.7 | 67.55 M | 0.57 s |
| PointPillars [9] | 77.3 | 52.3 | 62.7 | 64.1 | 4.83 M | 0.05 s |
| SECOND [8] | 78.6 | 53.0 | 67.2 | 66.3 | 5.32 M | 0.06 s |
| PointRCNN [61] | 78.7 | 54.4 | 72.1 | 68.4 | 4.09 M | 0.10 s |
| Part-A2 | 78.7 | **66.0** | **74.3** | 73.0 | 59.23 M | 0.14 s |
| PV-RCNN [7] | 83.6 | 57.9 | 70.5 | 70.7 | 13.12 M | 0.16 s |
| VoxelR-CNN [12] | 84.5 | - | - | - | 7.59 M | 0.05 s |
| DSTrans (w/o DCFT) | **84.8** | 62.7 | 73.1 | **73.5** | 15.23 M | 0.18 s |

Figure 10 presents the detection results in two different scenarios from the KITTI dataset. It can be observed that in the scenarios of enclosed roads and intersections, PV-RCNN tends to misclassify roadside trees and traffic facilities as pedestrians due to the lack of utilization of multi-scale information. In contrast, the proposed method, integrating multi-scale features, significantly reduces false positive objects. Without the addition of the DCST module, the false positive objects of PV-RCNN are mostly small pedestrian objects. This is attributed to the fact that voxel features obtained through 3D sparse convolution focus more on global features, leading to the loss of some local fine-grained details. The DCST module facilitates the exchange and fusion of multi-scale features, achieving more selective feature fusion based on deformable mechanisms.

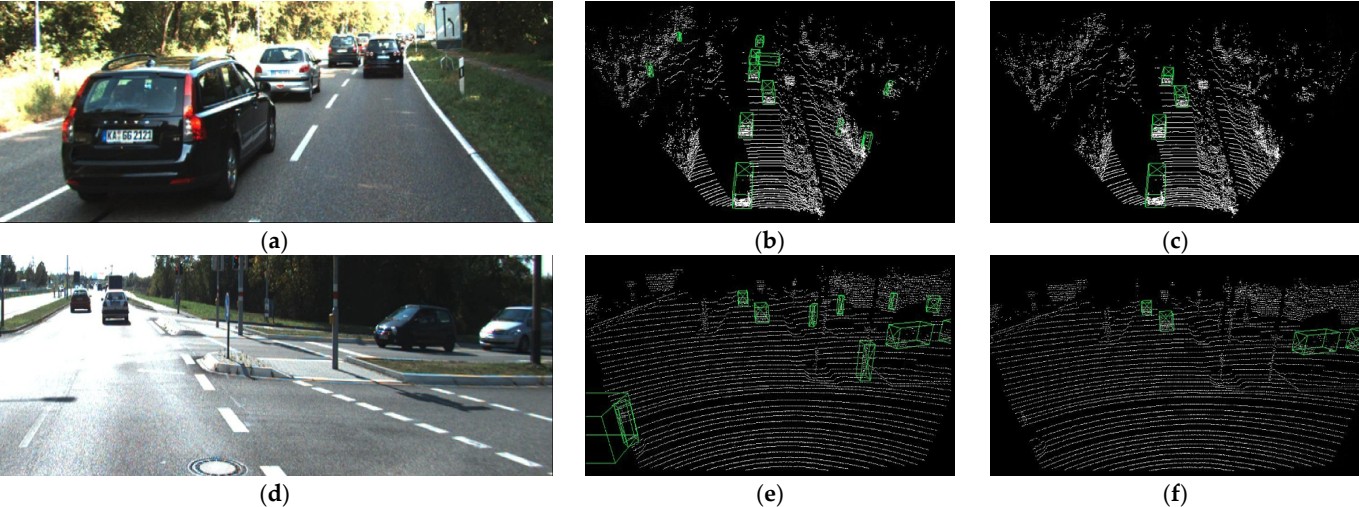

| (a) | (b) | (c) |
|---|---|---|

| (d) | (e) | (f) |
|---|---|---|

**Figure 10.** Visualization results of KITTI dataset. (**a**) The image of an enclosed road; (**b**) The detection results of PV-RCNN for an enclosed road; (**c**) The detection results of proposed method for an enclosed road; (**d**) The image of an intersection; (**e**) The detection results of PV-RCNN for an intersection; (**f**) The detection results of the proposed method for an intersection. The green boxes represent the predicted boxes by algorithm.

### 4.4. Results of the Self-Collected Dataset

To further validate the effectiveness of the proposed method in this paper, we constructed a small-scale dataset based on a real-world vehicle collection platform, specifically targeting urban road scenes in China. While the nuScenes and KITTI datasets provide extensive urban scene data, these datasets differ from Chinese urban roads in terms of background feature distribution and object density within the scenes. Figure 11 illustrates three common and complex urban scenarios in China. In the "Intersection Scene" in Figure 11a, the proposed method addresses the issue of missing vehicle object features due to occlusion

by fusing information from multiple frames, resulting in correct detections for the majority of objects. Similarly, in Figure 11c, vehicles parked alongside the road and partially obscured are also detected by proposed method. In Figure 11b, pedestrians, vehicles, and motorcycles are correctly detected for the majority of occurances. These results affirm the effectiveness of the proposed DSTrans in practical road environments.

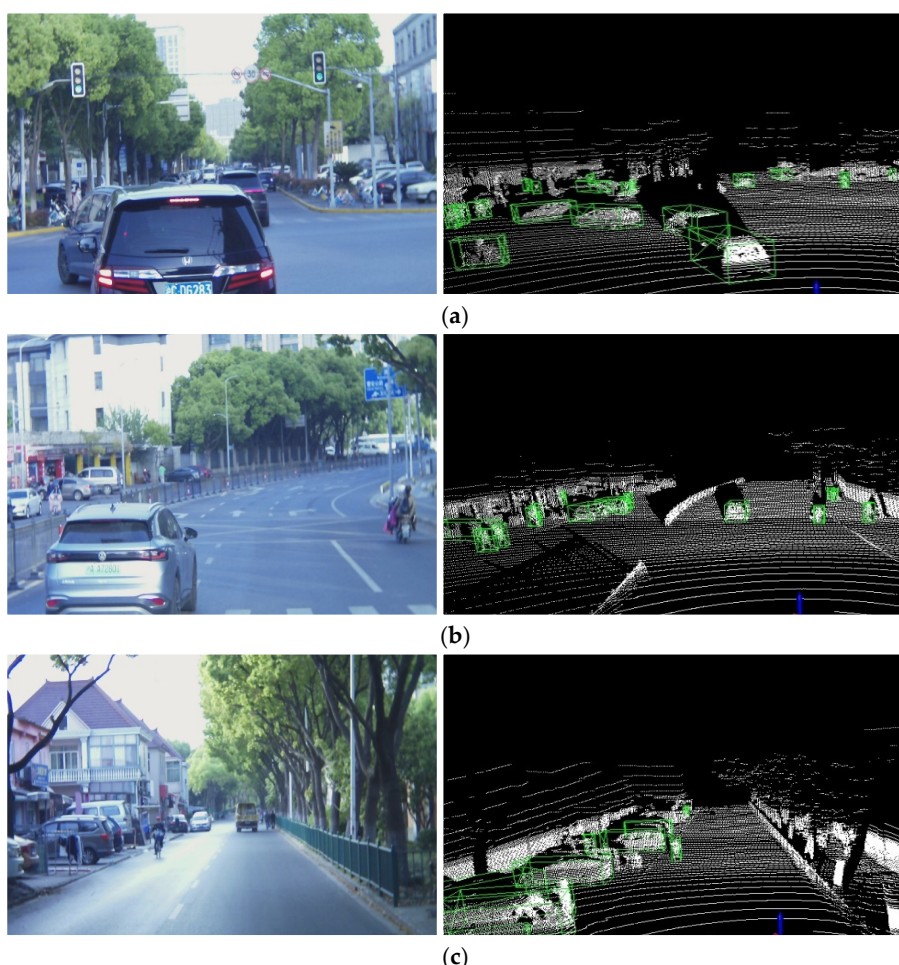

**Figure 11.** Visualization results of the self-collected dataset. (**a**) The detection results of the proposed method for an intersection; (**b**) The detection results of the proposed method for a main road; (**c**) The detection results of the proposed method for a suburban road. The green boxes represent the predicted boxes by algorithm.

*4.5. Ablation Studies*

4.5.1. Ablation Experiments of the DCST Module

The proposed DCST module facilitates the fusion of inter-frame voxel features across multiple scales, enhancing the representation capability for objects at various scales. To investigate the effectiveness of this module, several ablation experiments were designed to compare changes in detection results. As shown in Table 3, $F_1^s$, $F_2^s$, $F_3^s$, $F_4^s$ denote the application of deformable attention mechanisms only to the 1st, 2nd, 3rd, and 4th layer of voxel features. It can be observed that, after the fusion of features in the 4th layer, the performance improvement is most significant for the "Car" category, while the improvement for the "Pedestrian" and "Bicycle" categories is not obvious. This is because deep-level features tend to represent small objects in a coarse manner. However, by applying the DCST module with deformable attention mechanisms at each layer, the proposed method achieves the highest results across all categories and "mAP". The results in Table 3 demonstrate

the effectiveness of DCST in integrating multi-scale information, resulting in a noticeable enhancement in 3D object detection performance.

**Table 3.** Comparison of 3D object detection results for different scale fusion methods on the KITTI (moderate) validation dataset (%). The best results are marked in bold. The '$\sqrt{}$' marks indicate that the corresponding scale feature is utilized.

| $F_1^s$ | $F_2^s$ | $F_3^s$ | $F_4^s$ | Car | Pedestrian | Bicycle | mAP |
|---|---|---|---|---|---|---|---|
| $\sqrt{}$ | | | | 81.7 | 58.2 | 69.7 | 69.9 |
| | $\sqrt{}$ | | | 82.9 | 57.3 | 70.2 | 70.1 |
| | | $\sqrt{}$ | | 82.3 | 59.6 | 71.3 | 71.1 |
| | | | $\sqrt{}$ | 83.2 | 58.7 | 68.9 | 70.3 |
| $\sqrt{}$ | $\sqrt{}$ | $\sqrt{}$ | $\sqrt{}$ | **84.8** | **62.7** | **73.1** | **73.5** |

4.5.2. Ablation Experiments of the DCFT Module

To validate the effectiveness of the proposed DCFT for multi-frame feature integration, a comparison was conducted with results obtained using different numbers of fusion frames. As evident from the last four rows in Table 4, the 3D detection performance gradually improves with an increase in the number of fusion frames. The increment "$\Delta$" column in the table represents the detection performance improvement over the previous set of Settings.

**Table 4.** Comparison of 3D object detection results with different fusion frame numbers on the nuScenes validation dataset (%). Abbreviations: fusion frame numbers (FFN), construction vehicle (CV), pedestrian (Ped.), motorcycle (Motor.), and traffic cone (TC).

| Method | FFN | Car | Ped. | Bus | Barrier | TC | Truck | Trailer | Motor. | CV | Bicycle | mAP | $\Delta$ |
|---|---|---|---|---|---|---|---|---|---|---|---|---|---|
| PointPillars [9] | 1 | 68.4 | 59.7 | 28.2 | 38.9 | 30.8 | 23.0 | 23.4 | 27.4 | 4.1 | 1.1 | 30.5 | - |
| | 2 | 69.2 | 60.9 | 30.0 | 39.3 | 32.1 | 24.8 | 24.6 | 29.2 | 5.2 | 2.3 | 31.8 | 1.3 |
| | 3 | 70.1 | 60.4 | 30.7 | 38.7 | 31.4 | 22.1 | 23.8 | 30.5 | 5.7 | 2.9 | 31.6 | −0.1 |
| | 4 | 70.8 | 59.1 | 29.6 | 35.2 | 27.3 | 25.6 | 22.9 | 28.2 | 4.8 | 2.5 | 30.6 | −1.0 |
| DSTrans | 1 | 79.4 | 69.2 | 58.6 | 47.1 | 45.3 | 48.4 | 28.9 | 50.2 | 15.4 | 18.9 | 46.1 | - |
| | 2 | 81.5 | 72.7 | 59.1 | 50.4 | 48.7 | 51.3 | 29.6 | 53.4 | 17.2 | 23.4 | 48.7 | 2.6 |
| | 3 | 83.1 | 74.8 | 60.4 | 51.9 | 52.3 | 53.8 | 31.4 | 54.7 | 19.2 | 26.3 | 50.8 | 2.1 |
| | 4 | 84.2 | 77.3 | 60.9 | 52.7 | 52.9 | 54.2 | 32.6 | 55.2 | 19.8 | 27.2 | 51.7 | 0.9 |

It can be observed that employing DCFT for fusing adjacent two frames achieves the maximum improvement in the detection results for DSTrans. In contrast, Table 4 also analyzes the results of PointPillars using a method of overlaying multiple frames of raw data. While overlaying two frames of point cloud yields better results, overlaying three and four frames of data negatively impacts the detection performance. This suggests that the direct overlaying method does not effectively enhance the data representation of object features. The inconsistency in the positions of moving objects across multiple frames leads to distortion in the object features when directly overlaid. This result further emphasizes the effectiveness and necessity of the proposed DCFT.

However, it can be found that with the increase of FFN, the detection performance has not increased significantly from Table 4. This trend is gradually slowing down. Figure 12 shows the location and feature differences of different objects in the point cloud of four consecutive frames. It can be seen that there are differences of point clouds features and density distribution between objects at different times. When such differences are too large, more data is needed to train the model to adapt to the diversity and complexity brought by higher FFN. The results provide a promising direction for future research to explore advanced approaches that can more effectively deal with high variability across frames.

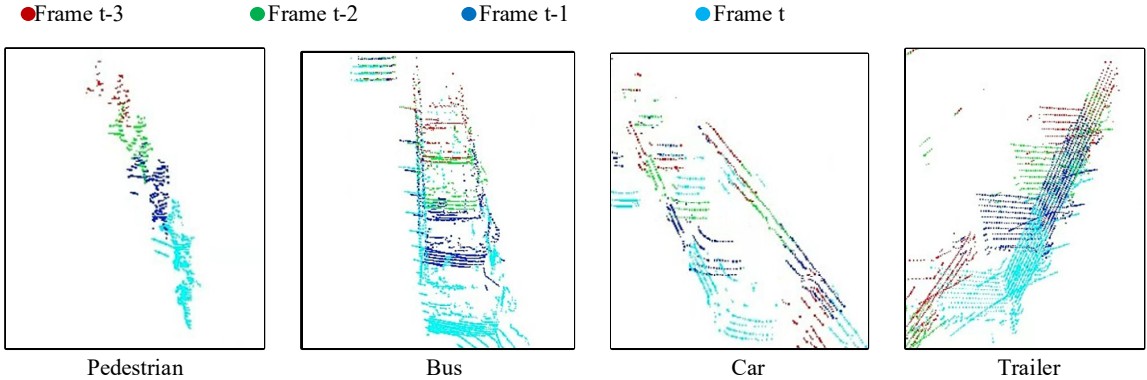

**Figure 12.** Different objects in the point cloud of four consecutive frames.

4.5.3. Ablation Experiments of the PAS Module

The choice of different sampling methods from the original point cloud can influence the aggregation of voxel features for key points. To investigate the impact of various sampling methods, while keeping the proposed model structure unchanged, key points were obtained using FPS, SPC, and PAS methods. These methods were tested on the nuScenes dataset.

FPS involves averaging sampling from the original point cloud, maintaining the point cloud density for each part while reducing the overall quantity. SPC divides a spherical region with the longest side of each proposal box as the radius to filter the object-related point cloud. The proposed PAS method simultaneously extracts proposals from two BEV features obtained from intra-frame and inter-frame voxel features, directly filtering the point cloud using the square region of the proposal box. PAS further improves the recall rate of foreground points based on SPC. Results presented in Table 5 demonstrate a significant improvement of 7.5% and 10.2% for SPC and PAS over the FPS method, respectively. This indicates that the proposed PAS module has a favorable impact on the performance of 3D object detection.

**Table 5.** Comparison of 3D object detection results with different sampling methods on the nuScenes validation dataset (%). Abbreviations: construction vehicle (CV), pedestrian (Ped.), motorcycle (Motor.), and traffic cone (TC).

| Method | Car | Ped. | Bus | Barrier | TC | Truck | Trailer | Motor. | CV | Bicycle | mAP |
|--------|-----|------|-----|---------|-----|-------|---------|--------|-----|---------|-----|
| FPS | 81.7 | 72.1 | 58.6 | 45.4 | 44.6 | 48.3 | 30.2 | 49.8 | 14.7 | 23.9 | 46.9 |
| SPC | 83.6 | 76.4 | 58.9 | 51.3 | 50.7 | 53.8 | 33.7 | 53.9 | 16.5 | 25.3 | 50.4 |
| PAS | 84.2 | 77.3 | 60.9 | 52.7 | 52.9 | 54.2 | 32.6 | 55.2 | 19.8 | 27.2 | 51.7 |

4.5.4. Ablation Experiments of the HVSA Module

Upon obtaining intra-frame voxel features $F_j^{\text{intra}}$ and inter-frame voxel features $F_j^{\text{inter}}$, it is essential to devise an appropriate fusion method to effectively integrate the information contained in both features into a key point. One direct approach involves element-wise addition of corresponding channels, followed by averaging to derive the key point feature. This process can be expressed as follows:

$$F_j^{keypoint} = \left( F_j^{\text{intra}} \oplus F_j^{\text{inter}} \right) / 2 \tag{19}$$

Table 6 compares the detection results between direct concatenation (Concat) and the proposed Hybrid Voxel Set Aggregation (HVSA). It is evident that HVSA outperforms direct concatenation, with the most significant improvements observed in the "Barrier" and "Traffic cone" categories. This indicates that the fusion method based on predictable weights proposed in this paper is effective in enhancing the performance of 3D object detection.

**Table 6.** Comparison of 3D object detection results with different feature fusion methods on the nuScenes validation dataset (%). Abbreviations: construction vehicle (CV), pedestrian (Ped.), motorcycle (Motor.), and traffic cone (TC).

| Method | Car | Ped. | Bus | Barrier | TC | Truck | Trailer | Motor. | CV | Bicycle | mAP |
|--------|-----|------|-----|---------|-----|-------|---------|--------|-----|---------|-----|
| Concat | 82.6 | 75.8 | 59.3 | 50.4 | 49.3 | 52.8 | 31.2 | 53.9 | 18.2 | 26.1 | 50.0 |
| HVSA | 84.2 | 77.3 | 60.9 | 52.7 | 52.9 | 54.2 | 32.6 | 55.2 | 19.8 | 27.2 | 51.7 |
| Increment | 1.6 | 1.5 | 1.6 | 2.3 | 3.6 | 1.4 | 1.4 | 1.3 | 1.6 | 1.1 | 1.7 |

## 5. Conclusions

To fully exploit the semantic correlations and spatiotemporal continuity inherent in temporal information, the proposed method specifically addresses the challenge of maintaining consistent perception results in the face of temporal instability or abrupt changes, particularly under complex weather conditions and on challenging road environments. This paper proposes a multi-frame 3D object detection algorithm based on the deformable spatiotemporal Transformer. Firstly, a Deformable Cross-Scale Transformer module is introduced to enhance the model's aggregation capability of spatial features. The Deformable Cross-Frame Transformer module is proposed to adaptively correlate dynamic features across multiple frames, improving the model's utilization of temporal information. Secondly, a proposal-aware sampling algorithm is designed to further optimize the efficiency of feature extraction. Finally, the obtained multiscale and multi-frame voxel features are adaptively fused using the proposed Hybrid Voxel Set Aggregation module, enabling the model to adaptively obtain fused features containing spatiotemporal information. Multiple experiments and ablation analyses conducted on the KITTI, nuScenes, and self-collected datasets validate the effectiveness of the proposed method. Future work will focus on optimizing implementation methods to reduce computational resource requirements for multi-frame fusion.

**Author Contributions:** Conceptualization, Y.Z., R.X. and H.W.; methodology, Y.Z., H.A. and R.X.; formal analysis, H.A.; supervision, K.L. and Z.S.; writing—original draft, R.X.; writing—review and editing, C.T. All authors have read and agreed to the published version of the manuscript.

**Funding:** This work was supported by the Perspective Study Funding of Nanchang Automotive Institute of Intelligence and New Energy, Tongji University under Grant TPD-TC202211-07.

**Data Availability Statement:** Publicly available datasets were analyzed in this study. These data can be found here: https://www.cvlibs.net/datasets/kitti/ (accessed on 15 January 2023) and www.nuscenes.org (accessed on 16 May 2023).

**Acknowledgments:** We appreciate the critical and constructive comments and suggestions from the reviewers that helped improve the quality of this manuscript. We also would like to offer our sincere thanks to those who participated in the data processing and provided constructive comments for this study.

**Conflicts of Interest:** The authors declare no conflicts of interest.

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
