# Peer review of "DS-Trans: A 3D Object Detection Method Based on a Deformable Spatiotemporal Transformer for Autonomous Vehicles"

_remotesensing, doi:10.3390/rs16091621_

Round 1

Reviewer 1 Report

Comments and Suggestions for Authors

See attachment.

Reviewer 2 Report

Comments and Suggestions for Authors

Dear authors, 

Thank you for your efforts in producing this paper. In general, the paper is well written, and I have only few questions or suggestions listed below:

In section 4.1.3 - I suggest providing a more detailed description of the specific sensors utilized in the measurement.

Section 4.2 (lines 417 – 421) – some of the listed categories are more easily detectable by your method, while some of them are less detectable, how you can explain it?

The processing part of the algorithm is done in real-time or post-processing? And also, the data before the processing is somehow pre-processed or it is used as “measured”? Perhaps implementing some pre-processing filters could improve measurements taken in rainy conditions by eliminating “false reflections”.

 Regards

Comments on the Quality of English Language

 Minor editing of English language required.

Reviewer 3 Report

Comments and Suggestions for Authors

The manuscript entitled “DS-Trans: An 3D Object Detection Method Based on Deformable Spatiotemporal Transformer for Autonomous Vehicle” is work that designed a new Transformer-based deep learning algorithm for 3D object detection from multi-frame point clouds. It performs better on the test data set than the other popular algorithms. The main contribution of this work is the investigation of the model structure, which is not significantly innovative but effective. My concerns are as follows:

1) It is recommended to compare the number of trainable parameters and computational complexity with other models.

2) L378, please double check the title of Figure 5.

3) L404, Section 4.1.3, please specify the number and distribution of self-collected samples and whether they are publicly available.

4) L562, Table V or Table 5? L572, Table 6 or Table VI? Please have a check.

5) L595, Table 4, it seems that with the increase of FFN, the performance of the proposed model increases, so why not continue to increase it, such as increasing it to 10, please further analyze their correlation.

Round 2

Reviewer 3 Report

Comments and Suggestions for Authors

After revision, the manuscript has been improved. The only issue I still concerned is the correlation between the model performance and its FFN. As explained in the cover letter, the authors should add such explanations or experiment results in the paper.

Author Response

Thank you for your suggestion. We have added more explanations into the 4.5.2 section.